# Enhanced Bacteremia in Dextran Sulfate-Induced Colitis in Splenectomy Mice Correlates with Gut Dysbiosis and LPS Tolerance

**DOI:** 10.3390/ijms23031676

**Published:** 2022-01-31

**Authors:** Arthid Thim-Uam, Jiradej Makjaroen, Jiraphorn Issara-Amphorn, Wilasinee Saisorn, Dhammika Leshan Wannigama, Wiwat Chancharoenthana, Asada Leelahavanichkul

**Affiliations:** 1Division of Biochemistry, School of Medical Sciences, University of Phayao, Phayao 56000, Thailand; arthidth@gmail.com; 2Center of Excellence in Systems Biology, Faculty of Medicine, Chulalongkorn University, Bangkok 10400, Thailand; jiradejmak@gmail.com; 3Department of Microbiology, Faculty of Medicine, Chulalongkorn University, Bangkok 10400, Thailand; jiraphorn298@gmail.com (J.I.-A.); wsaisorn@gmail.com (W.S.); leshanwannigama@gmail.com (D.L.W.); 4Antimicrobial Resistance and Stewardship Research Unit, Faculty of Medicine, Chulalongkorn University, Bangkok 10400, Thailand; 5School of Medicine, Faculty of Health and Medical Sciences, The University of Western Australia, Nedlands, WA 6009, Australia; 6Tropical Nephrology Research Unit, Department of Clinical Tropical Medicine, Faculty of Tropical Medicine, Mahidol University, Bangkok 10400, Thailand; wiwat.cha@mahidol.ac.th; 7Tropical Immunology and Translational Research Unit, Department of Clinical Tropical Medicine, Faculty of Tropical Medicine, Mahidol University, Bangkok 10400, Thailand; 8Translational Research in Inflammation and Immunology Research Unit (TRIRU), Department of Microbiology, Chulalongkorn University, Bangkok 10400, Thailand; 9Division of Nephrology, Department of Medicine, Faculty of Medicine, Chulalongkorn University, Bangkok 10400, Thailand

**Keywords:** splenectomy, gut barrier defect, endotoxemia, dysbiosis

## Abstract

Because both endotoxemia and gut dysbiosis post-splenectomy might be associated with systemic infection, the susceptibility against infection was tested by dextran sulfate solution (DSS)-induced colitis and lipopolysaccharide (LPS) injection models in splenectomy mice with macrophage experiments. Here, splenectomy induced a gut barrier defect (FITC-dextran assay, endotoxemia, bacteria in mesenteric lymph nodes, and the loss of enterocyte tight junction) and gut dysbiosis (increased Proteobacteria by fecal microbiome analysis) without systemic inflammation (serum IL-6). In parallel, DSS induced more severe mucositis in splenectomy mice than sham-DSS mice, as indicated by mortality, stool consistency, gut barrier defect, serum cytokines, and blood bacterial burdens. The presence of green fluorescent-producing (GFP) *E. coli* in the spleen of sham-DSS mice after an oral gavage supported a crucial role of the spleen in the control of bacteria from gut translocation. Additionally, LPS administration in splenectomy mice induced lower serum cytokines (TNF-α and IL-6) than LPS-administered sham mice, perhaps due to LPS tolerance from pre-existing post-splenectomy endotoxemia. In macrophages, LPS tolerance (sequential LPS stimulation) demonstrated lower cell activities than the single LPS stimulation, as indicated by the reduction in supernatant cytokines, pro-inflammatory genes (*iNOS* and *IL-1**β*), cell energy status (extracellular flux analysis), and enzymes of the glycolysis pathway (proteomic analysis). In conclusion, a gut barrier defect after splenectomy was vulnerable to enterocyte injury (such as DSS), which caused severe bacteremia due to defects in microbial control (asplenia) and endotoxemia-induced LPS tolerance. Hence, gut dysbiosis and gut bacterial translocation in patients with a splenectomy might be associated with systemic infection, and gut-barrier monitoring or intestinal tight-junction strengthening may be useful.

## 1. Introduction

As the largest lymphatic organ, the spleen’s functions are filtration of foreign matters from blood, antibody production, and bacteria control [1]. Removal of the spleen (splenectomy), mostly due to blunt splenic injury [2], hypersplenism [3,4], and functional hyposplenism [5,6,7] in some conditions (coeliac disease, inflammatory bowel disease, and sickle cell anemia) [8] can lead to several bacterial infections, including *Klebsiella pneumoniae*, *Hemophilus influenzae,* and *Pseudomonas aeruginosa* [5,6,7]. Although these bacteria could be found in the environment, they are encapsulated Gram-negative bacteria that could be found in the intestine [9]. In addition, the detection of lipopolysaccharides (LPS) (a major component of Gram-negative bacterial cell walls) in serum (endotoxemia) in splenectomy patients from abdominal trauma [10] and thalassemia [11] and the association between splenic denervation and LPS responses [12,13,14] are also reported. These data indicate a possible spleen–gut correlation, despite the limited direct exploration of the intestinal epithelium after splenectomy. Our objective was to demonstrate and explore the clinical impact of an intestinal barrier defect post-splenectomy that might facilitate spontaneous systemic infection.

As such, epithelial tight junctions hold enterocytes together, forming the first phase of the intrinsic mucosal defense system as a selective physical barrier between the host and gut contents [15]. Increased gut permeability (gut barrier defect or leaky gut) causes augmentation on the translocation of viable pathogens, referred to as “gut translocation or gut bacterial translocation”, or pathogen molecules, including LPS, through the gut wall into blood circulation [16,17]. Intestinal inflammation or severe systemic inflammation can cause leaky gut through the direct damage on tight junctions (such as hypoxia and inflammatory cells) and/or indirect injuries from enhanced intestinal pathogenic organisms (gut dysbiosis) [18,19]. Beneficial bacteria in the gut are important for normal gut integrity due to several factors, including (i) the production of short-chain fatty acids (an energy source of enterocytes) by intestinal fermentation of indigestible foods [20] and (ii) the regulation of pathogenic bacteria, partly through competition for nutrients [21]. As a result, gut dysbiosis can lead to a gut barrier defect [15]. Subsequently, the significant breach of the gut barrier can exacerbate or cause the translocation of pathogen molecules from the gut into blood circulation that facilitates immune responses against these molecules, including LPS, causing more severe systemic inflammation [22,23]. Among immune cells, macrophages are responsible for the LPS responses due to several pattern recognition receptors, especially toll-like receptor 4 (TLR-4), on the cell surface [24], which might be responsible for characteristics of endotoxemia in mice. Despite severe sepsis from hyper-cytokine production after a single LPS stimulation on macrophages, the repeated LPS stimulation, perhaps from chronic endotoxemia, might induce less pro-inflammatory cytokines referred to as “LPS tolerance”, leading to insufficient cytokines for organismal control (chemotaxis and enhanced cell activities) [25] that might be found during sepsis immune exhaustion (the post-sepsis condition that is susceptible to secondary infections due to inadequate responses against pathogens) [26,27].

Although a gut barrier defect after splenectomy is not severe enough to cause gastrointestinal symptoms (such as diarrhea), the pre-existing intestinal injury might make it more susceptible to additional insults, similar to several two-hit injury models [22,23]. To test this hypothesis, dextran sulfate solution (DSS), which induces direct damages on enterocytes [11], was administered in splenectomy mice and sham mice. Likewise, a chronic inflammatory state after splenectomy might be associated with more severe systemic infection due to the exhaustion to LPS stimulation (LPS tolerance) [25]. Then, the immune responses against LPS (single and sequential doses) were also tested in these mice with several experiments on macrophage LPS tolerance. Hence, tests on gut-barrier function and fecal dysbiosis in splenectomy mice with or without DSS-induced mucositis and LPS administration models with the in vitro experiments on macrophages were conducted.

## 2. Results

### 2.1. Splenectomy-Induced Gut Barrier Defect

The splenectomy-induced gut barrier defect was indicated by FITC-dextran (4.4 kDa) assay, endotoxemia from gut translocation, and positive bacterial culture in the mesenteric lymph nodes at 14 days post-splenectomy (Figure 1A–C). However, leaky gut was not severe enough to detect blood bacteria (non-bacteremia) (data not shown). In comparison with the sham, there was a lower fluorescent intensity of intestinal tight-junction molecules (occludin and ZO-1) (at 14 days) and more severe liver damage (alanine transaminase) (at 3 days) in post-splenectomy mice without the difference in serum IL-6 and serum creatinine (a kidney function biomarker) (Figure 1D–H).

### 2.2. The Enhanced Dextran Sulfate Solution-Induced Mucositis Due to Splenectomy-Induced Gut Barrier Defect

Although the gut barrier defect from splenectomy was not severe enough to demonstrate the clinical significance (diarrhea, serum cytokines, and organ injuries) at 14 days post-splenectomy (Figure 1F–H), the intestinal defect made it more susceptible to the injury by dextran sulfate solution (DSS) that was administered at 14 days post-splenectomy (Figure 2A).

Accordingly, DSS was a cause of mortality only in splenectomy mice, but not in sham mice, and induced more severe diarrhea as indicated by the earlier diarrheal onset (stool consistency index), despite the non-different weight loss (Figure 2B–D). At 7 days post-DSS, the colon histology score (as determined by leukocyte infiltration, enterocyte injury, and ulceration) and gut barrier defect (endotoxemia and FITC-dextran assay) in splenectomy-DSS were more severe than sham-DSS (Figure 2E–G and Figure 3). Leaky gut in splenectomy mice induced higher serum cytokines (TNF-α, IL-6, and IL-10) (Figure 2H–J) and bacteremia (Figure 4A) than sham mice. The identified bacterial colonies were *Staphylococcus coagulase* negative and *Pseudomonas aeruginosa* in sham-DSS, while they were identified as pathogenic bacteria (such as *Klebsiella pneumoniae* and *P. aeruginosa*) in splenectomy-DSS mice (Figure 4A, the list of bacteria). Because these bacteria in blood might be transferred from the intestine (gut barrier defect), GFP *E. coli* were orally administered at 6 h before sacrifice in DSS and water mice. Without DSS, the GFP bacteria were demonstrated only in the mesenteric lymph nodes of splenectomy mice (Figure 4B,C) but not in the lymph nodes of sham mice (data not shown). In sham-DSS mice, GFP bacteria were demonstrated in all interested organs (mesenteric lymph nodes, livers, kidneys, and spleens) with the highest abundance in the spleen (Figure 4B,C). In splenectomy-DSS mice, the fluorescent intensities of the GFP bacteria in all organs (except for the spleens) were higher than sham-DSS (most prominent in the livers) (Figure 4B,C).

### 2.3. Gut Dysbiosis in Splenectomy Mice with and without Dextran Sulfate Solution (DSS)

Fecal microbiome analysis was performed at 3 weeks post-splenectomy (1 week after DSS). In the water control group, splenectomy-induced gut dysbiosis when compared with sham mice as indicated by increased Proteobacteria, especially *E. coli* and *Shigella* (pathogenic bacteria), decreased *Alistipes* (Bacteroides bacteria), and reduced microbial diversity without a difference in total fecal Gram-negative bacteria (Figure 5A–F). After DSS administration, splenectomy induced several differences from sham-DSS mice, as indicated by (i) higher *E. coli, Shigella*, and *Alistipes* in the genus-level analysis, despite the lower Proteobacteria (in phylum-level analysis), (ii) higher Verrucomicrobia and *Akkermansia*, (iii) lower *Allobacum* and *Ruminococcus*, and (iv) lower total Gram-negative bacteria and bacterial diversity (Figure 5C–F). Despite the lower total Gram-negative bacteria in the feces of splenectomy-DSS mice compared with sham-DSS mice (Figure 5E), endotoxemia in splenectomy-DSS mice was more severe than sham-DSS mice (Figure 2E).

### 2.4. Chronic Endotoxemia in Splenectomy Mice Possibly Resulted in the Lower Serum Cytokines in Both Single and Sequential LPS Administration

To test responses against LPS, the injection by a single dose (LPS pro-inflammation) or sequential LPS protocol (LPS/LPS or LPS tolerance) was performed (Figure 6A). Interestingly, splenectomy reduced inflammatory cytokines (TNF-α and IL-6 but not IL-10) at 15 min after LPS administration (Figure 6B–G). Although the development of LPS tolerance (reduction of cytokine responses after the first dose of LPS) was similar between sham and splenectomy mice, levels of the cytokines after LPS/LPS stimulation in splenectomy mice (at 15 and 30 min post-injection) were lower than sham mice (Figure 6B–G). Serum cytokines after a single LPS administration in splenectomy mice were lower than LPS-sham mice (partly from the loss of splenic cytokine production) (Figure 6B–G). In parallel, serum cytokines in splenectomy-DSS mice were higher than sham-DSS (Figure 2J–L) because of the higher burdens of blood bacteria (Figure 4A) that possibly induced cytokine production from other organs.

### 2.5. Macrophage Endotoxin Tolerance, a Possible Impact from Chronic Endotoxemia in Splenectomy Mice

Chronic endotoxemia in the splenectomy mice (Figure 1A and Figure 2E) might be responsible for inadequate cytokines for infection control leading to profound bacteremia (Figure 4A) and severe DSS-induced sepsis (Figure 2B). Here, the characteristics of LPS tolerance of macrophages, in comparison with a single LPS response, (Figure 7A) were demonstrated through the reduction of supernatant cytokines, downregulated pro-inflammatory genes (*iNOS* and *IL-1β*), and upregulated anti-inflammatory genes (*Arginase-1* and *TGF-β*) (Figure 7B–H).

Additionally, LPS tolerance reduced the cell energy status in both mitochondrial and glycolysis stress tests, while the single LPS stimulation enhanced the glycolysis stress test, as indicated by the maximal glycolysis and maximal respiratory capacity when compared with the control group (Figure 7I–L). The proteomic analysis of the LPS tolerance macrophage (sequential LPS stimulation) relative to the single LPS stimulation totally demonstrated 2444 proteins with significantly 565 upregulated and 800 downregulated proteins, respectively. Indeed, LPS tolerance reduced the energy from glycolysis, as indicated by the decreased phosphofructokinase (Pfk; a rate-limiting glycolysis enzyme) but enhanced the glucose utilization by hexokinase (Hk) (Figure 8). Meanwhile, LPS tolerance upregulated several enzymes of the mitochondrial pathway, including pyruvate kinase (Pkm) (Figure 8), possibly to maximize the mitochondrial activity.

## 3. Discussion

Post-splenectomy leaky gut, dysbiosis, and endotoxemia enhanced the bacteremia of DSS-induced mucositis, implying the impact of the loss of microbial control due to asplenia and macrophage LPS tolerance.

### 3.1. Splenectomy-Induced Dysbiosis, Gut Barrier Defect, and Gut Trsnslocation

The spleen is vital for microbial control, especially of bloodborne organisms [28]. In healthy persons, transient bacteremia from gut translocation is a possible natural event that occurs on a regular basis with no harmful consequences [29] because of intact spleen functions. Without a spleen, transient bacteremia can develop into a more serious infection [30,31]. In post-splenectomy without DSS, endotoxemia (without bacteremia) might be a consequence of leaky gut, as indicated by the FITC-dextran (4.4 kDa) assay, viable bacteria in mesenteric lymph nodes, and loss of intestinal tight-junction molecules, which was severe enough for the translocation of LPS (50–100 kDa) but not bacteria (higher than 1 × 10^6^ kDa) [15,32]. Although the direct communication between the intestine and the spleen is not clear, splenectomy facilitated gut dysbiosis [10,33] (partly from stress-induced dysbiosis [34]), as indicated by increased pathogenic Proteobacteria (*E. coli* and *Shigella*) with decreased bacterial diversity [33,35]. The stress from endotoxemia and gut dysbiosis, referred to as the “gut–brain axis” [34,36,37,38], might subsequently worsen leaky gut [39,40] and further induces a higher level of endotoxemia as a vicious cycle [41,42,43] (Figure 9, in the middle). Although all of the major mechanisms of gut translocation-induced bacteremia, including gut dysbiosis, gut barrier defect, and host immune deficiencies [44], were demonstrated in splenectomy mice, these defects resulted in endotoxemia but were not severe enough to cause bacteremia.

### 3.2. Post-Splenectomy Bacteremia of Dextran Sulfate-Induced Mucositis Mice, the Loss of Microbial Control from Asplenia, and LPS Tolerance

The spleen’s impact on the control of bacteria from gut translocation was demonstrated by the highest fluorescent intensity of GFP *E. coli* in the spleens of sham-DSS mice. In splenectomy-DSS, the intensity in all organs was higher than the organs in sham-DSS, indicating the compensation of bacterial dissemination during asplenia. Because of pre-existing splenectomy-induced leaky gut, post-splenectomy mice were more susceptible to DSS-induced mucositis and gut translocation, as bacteremia presented in 20% of the sham-DSS mice (mostly *Staphylococcus* coagulase negative bacteria) and 80% of splenectomy-DSS mice (mostly encapsulated *Klebsiella pneumoniae*). In parallel, gut dysbiosis in splenectomy-DSS mice was also more prominent than sham-DSS, as *E. coli* and *Shigella* (pathogenic bacteria) and *Alistipes* (Bacteroides bacteria possibly causing inflammation [48]) were more prominent in splenectomy-DSS mice, despite the higher beneficial bacteria (Verrucomicrobia [49] and *Akkermansia* [50]) with lower pro-inflammatory bacteria (*Allobacum* and *Ruminococcus* [51]). However, minor differences in intestinal dysbiosis between mice with versus without DSS (both sham and splenectomy) suggested that DSS-induced dysbiosis was less important than the direct damage from DSS in causing endotoxemia.

On the other hand, chronic endotoxemia or repeated endotoxin exposure post-splenectomy might induce an LPS tolerance [25,52] that inadequately induces pro-inflammatory cytokines for the proper microbial control [25,52]. In a clinical situation, patients with splenectomy develop limited immune responses to infection, and low-grade fever in these patients is an indication for aggressive investigations with early antibiotic administration [53]. Here, splenectomy mildly reduced serum cytokines in an early phase of single and sequential LPS, supporting the spleen’s impact on (i) systemic cytokine production [54,55,56,57], (ii) a source of cytokines [57,58,59], and (iii) post-splenectomy anti-inflammation [60,61,62]. Notably, post-splenectomy anti-inflammation could be beneficial or harmful toward sepsis, as hyper-inflammation induces sepsis death [27] while inadequate inflammation causes profound bacterial burdens and mortality [25,52]. Hence, we hypothesized that prominent bacteremia in splenectomy-DSS might partly be because of relatively inadequate immune responses against the early phase of bacteremia (Figure 9, the right figure). Meanwhile, the higher serum cytokines in splenectomy-DSS over sham-DSS mice might be due to the higher bacterial burdens in splenectomy-DSS mice.

### 3.3. LPS Tolerance Macrophage, Possibly Enhanced Bacteremia in Dextran Sulfate-Induced Endotoxemia with Splenectomy

Although the loss of microbial control due to asplenia is extensively mentioned, studies on the possible impact of LPS tolerance (due to chronic endotoxemia post-splenectomy) on organismal control by macrophages [63,64] are still less. As such, LPS tolerance of macrophages induced lower cytokines, upregulated anti-inflammatory genes (*Arginase-1* and *TGF-β*), downregulated pro-inflammatory genes (*iNOS* and *IL-1β*), and reduced cell energy status (both mitochondria and glycolysis) when compared with the single LPS stimulation [65,66,67]. With proteomic analysis, LPS tolerance reduced enzymes for glycolysis energy production, a major energy source for macrophage cytokine production [68], as indicated by decreased phosphofructokinase (a rate-limiting enzyme of glycolysis), which was compensated by enhanced glucose utilization by increased hexokinase [69]. Because of the reduced mitochondrial activity by LPS tolerance (extracellular flux analysis), several mitochondrial enzymes (pyruvate kinases) were upregulated, possibly to maximize mitochondrial activity [70]. Because macrophages have pleiotropic functions, including pro- and anti-inflammatory properties [71,72], and interventions for the harnessing of macrophage functions into the proper direction are mentioned [73], post-splenectomy immunoadjuvant and/or gut-barrier monitoring might be beneficial. Precautions for severe infection, treatments for gut dysbiosis, and early antibiotics might be necessary for post-splenectomy individuals with endotoxemia or leaky gut, while immune interventions (enhanced macrophage activities) might be beneficial in these patients with severe infection. More studies are interesting.

In conclusion, post-splenectomy leaky gut was susceptible to superimposed intestinal injury by DSS, which induced more severe endotoxemia and bacteremia (due to loss of splenic microbial control and LPS tolerance), resulting in increased mortality. Hence, gut-barrier monitoring and tight-junction strengthening might be beneficial in patients with splenectomy.

## 4. Materials and Methods

### 4.1. Animals and Animal Models

The animal care and use protocol was certified by the Institutional Animal Care and Use Committee of Chulalongkorn University’s Faculty of Medicine in Bangkok, Thailand, in compliance with the U.S. National Institutes of Health criteria. Male 8-week-old C57BL/6 mice (purchased from Nomura Siam International, Pathumwan, Bangkok, Thailand) had free access to water and chow before and after surgery.

#### 4.1.1. Splenectomy and Dextran Sulfate Solution

Splenectomy was performed through a left flank incision, and the splenic pedicles were ligated using 4-0 silk sutures, as previously described [74,75]. In the sham group, a flank incision was made to identify the spleen before closing. Blood samples (50 µL) were collected through tail vein nicking in several time-points. In parallel, dextran sulfate solution (DSS) (Sigma-Aldrich, St. Louis, MO, USA) was diluted into the mouse drinking water at 2 weeks after surgery at a concentration of 3% volume by volume (*v*/*v*) for 1 week to induce gut barrier defect, following previous publications [17,76,77]. The stool consistency was semi-quantitatively evaluated using the following score: 0, normal; 1, soft; 2, loose; 3, diarrhea, as previously published [78].

#### 4.1.2. Single or Sequential LPS Administration in Splenectomy Mice

To test the inflammatory responses, splenectomy (or sham) mice after 2 weeks post-operation were intravenously (tail vein) administered with LPS in a single (sepsis) or sequential dose (LPS tolerance) [25,52,79,80]. For sequential LPS, LPS from *Escherichia coli* 026: B6 (Sigma-Aldrich) were administered in two separate doses at 0.8 and 4 mg/kg with 5-day rest between doses. For single LPS, phosphate buffer solution (PBS) was followed by 4 mg/kg LPS 5 days later. Then, twice PBS was used for the control group. Blood (50 µL) was collected through the facial artery in the indicated time-points. Serum cytokines and kidney injury (serum creatinine) were determined by enzyme-linked immunosorbent assays (ELISA) (Invitrogen, Carlsbad, CA, USA) and QuantiChrom (DICT-500; BioAssay, Hayward, CA, USA), respectively. Liver damage was determined by EnzyChrom alanine trans-aminase assay (EALT-100, BioAssay). Furthermore, blood (25 μL) was directly spread onto blood agar plates (Oxoid, Hampshire, UK), incubated at 37 °C for 24 h before bacterial colony enumeration, and colony identification was performed by mass spectrometry analysis by ViTek MS (BioMerieux SA, Marcy-l’Etoile, France), following the routine hospital protocol. At the end of experiments, mice were sacrificed with specimen collection through cardiac puncture under isoflurane anesthesia.

### 4.2. Measurement of Gut Barrier Defect

Gut permeability was determined by FITC-dextran assay [22,81,82], endotoxemia [15], fluorescent staining of tight-junction molecules, bacterial burdens in mesenteric lymph nodes [82,83], and dissemination of green fluorescent-producing (GFP) *E. coli* [11]. Accordingly, FITC-dextran, a nonabsorbable molecule with 4.4 kDa molecular mass (Sigma-Aldrich), at 12.5 mg was orally administered at 3 h before detection in serum using fluorospectrometer (NanoDrop 3300; ThermoFisher Scientific, Wilmington, DE, USA). In parallel, serum endotoxin (LPS) was measured by HEK-Blue LPS detection (InvivoGen, San Diego, CA, USA) and data were recorded as 0 when LPS values were less than 0.01 EU/mL because of the limited lower range of the standard curve. For fluorescent staining of intestinal tight-junction molecules, the colon samples were put into tissue frozen in OCT compound (Tissue-Tek OCT compound; Sukura Finetek, Inc., Torrance, CA, USA) prepared in 5 mm thick frozen tissue sections before processing with acetone and blocking reagents [22]. Subsequently, the primary antibodies against enterocyte tight-junction molecules of occludin or zonula occludens-1 (ZO-1) (Thermo Fisher Scientific, Bannockburn, IL, USA) (1:200) were incubated, washed, mixed with secondary antibody Alexa Fluor 546 goat anti-rabbit IgG (Life Technologies, Carlsbad, CA, USA) (1:200) at 23 °C for 1 h, visualized, and scored (fluorescent intensity) with a Zeiss LSM 800 confocal microscope (Carl Zeiss, San Diego, CA, USA). Because detection of viable bacteria in mesenteric lymph nodes is an indirect indicator of viable bacterial translocation through the intestine, microbial burdens in mesenteric lymph nodes were determined. As such, the homogenized lymph nodes in PBS were directly spread onto blood agar (Oxoid) and incubated at 37 °C for 24 h before colony enumeration. To further support gut translocation, GFP *E. coli* (25922GFP) from American Type Culture Collection (ATCC, Manassas, VA, USA) at 1 × 10^9^ CFUs in 0.3 mL PBS were orally administered at 6 h before sacrifice, and fluorescent intensity in internal organs was detected using ZEISS LSM 800 (Carl Zeiss), as mentioned above.

### 4.3. Colon Histology

The following scores were used to semi-quantitatively evaluate intestinal histology on hematoxylin and eosin (H&E) staining at 200× magnification based on mononuclear cell infiltration (in mucosa and submucosa), epithelial hyperplasia (epithelial cells in longitudinal crypts), reduction of goblet cells, and epithelial cell vacuolization in comparison to control with the following score: 0, no epithelial hyperplasia (control); 1, leukocyte infiltration 5–10% or hyperplasia 10–25%; 2, leukocyte infiltration 10–25% or hyperplasia 25–50% or reduced goblet cells (control); 3, leukocyte infiltration 25–50% or hyperplasia >50% or intestinal vacuolization; 4, leukocyte infiltration >50% or ulceration.

### 4.4. Fecal Microbiome Analysis

Feces from mice (0.25 g each mouse) were used for the analysis using an Illumina MiSeq at Omics Sciences and Bioinformatics Center (Chulalongkorn University, Bangkok, Thailand), as previously described [82,84,85]. Microbiome bioinformatics were performed with QIIME 2 2019.10 [86]. Raw sequence data were demultiplexed and quality filtered using the q2-demux plugin followed by denoising with DADA2 [87] (via q2-dada2). A phylogeny was constructed using SEPP q2-plugin, placing short sequences into sepp-refs-gg-13-8.qza reference phylogenetic tree for 16S marker gene. Taxonomy was assigned to ASVs using the q2-feature-classifier classify-sklearn naïve Bayes taxonomy classifier against the SILVA database. Beta diversity metrics using Chao-1 (richness estimation; the estimation on species diversity from the abundance of data) and Shannon (evenness analysis; the number of species present) were estimated using q2-diversity [88] after samples were rarefied (subsampled without replacement) to the minimum number of sequences.

### 4.5. Macrophage Cytokines and Polymerase Chain Reaction

To demonstrate the possible influence of chronic endotoxemia in mice through macrophage responses, activations by single or sequential doses of LPS, representatives for acute LPS inflammation [89], and LPS tolerance [25], respectively, were performed. As such, macrophages were derived from the bone marrow of 8-week-old mice, as previously described [81,84,90,91]. Briefly, bone marrow from femurs and tibias were collected by centrifugation at 6000 rpm for 4 °C and incubated for 7 days with Dulbecco’s Modified Eagle Medium (DMEM) supplemented with 10% fetal bovine serum (FBS), 1% penicillin/streptomycin, 4-(2-hydroxyethyl)-1-piperazineethanesulfonic acid (HEPES) with sodium pyruvate in a humidified 5% CO_2_ incubator at 37 °C. Conditioned media of the L929 cell line, containing macrophage-colony stimulating factor, at 20% weight by volume (*w*/*v*) were used to induce macrophages from the pluripotent stem cells. There were three stimulation protocols. For LPS tolerance (LPS/LPS), LPS (*Escherichia coli* 026: B6) (Sigma-Aldrich) at 100 µg/mL was incubated with macrophages at 5 × 10^5^ cells/well for 24 h before 6 h DMEM washing period, followed by 24 h LPS (100 µg/mL) incubation and sample collection (cells and supernatant). In single LPS stimulation (PBS/LPS), the protocol started with 24 h DMEM media followed by 6 h DMEM washing period before 24 h LPS incubation. In control group (media/media), only DMEM media, but not LPS, were used during the process. Subsequently, supernatant cytokines were measured by ELISA (Invitrogen), and cells were evaluated by real-time polymerase chain reaction (PCR), as previously published [89]. Briefly, an RNA easy mini kit (Qiagen, Hilden, Germany) and a high-capacity reverse transcription assay (Applied Biosystems, Warrington, UK) were used to prepare total RNA and reverse transcription, respectively. The comparative threshold (delta-delta Ct) approach (2^−ΔΔCt^) was used to establish relative quantification using an Applied Biosystems 7500 Real-Time PCR System with SYBR^®^ Green PCR Master Mix (Applied Biosystems) as normalized by Beta-2 microglobulin (β2M, an endogenous housekeeping gene). The primers are the following: inducible nitric oxide synthase (*iNOS*) forward 5′-ACCCACATCTGGCAGAATGA-3′ reverse 5′-AGCCATGACCTTTCGCATTAG-3′; interleukin-1 beta (*IL-1β*) forward 5′-GAAATGCCACCTTTTGACAGTG-3′ reverse 5′-TGGATGCTCTCATCAGGACAG-3′; arginase-1 forward 5′-CTTGGCTTGCTTCGGAACTC-3′ reverse 5′-GGAGAAGGCGTTTGCTTAGTT-3′; transforming growth factor-beta (*TGF-β*) forward 5′-CAGAGCTGCGCTTGCAGAG-3′ reverse 5′-GTCAGCAGCCGGTTACCAAG-3′; β2 microglobulin (*β2M*) forward 5′-CCACTGAAAAAGATGAGTATGCCT-3′ reverse 5′-CCAATCCAAATGCGGCATCTTCA-3′.

### 4.6. Extracellular Flux Analysis

Because cell energy status is important for LPS-induced cytokine production and other cell activities [79], cell energy in single and sequential LPS stimulated macrophages was evaluated. As such, the energy metabolism profiles, using Seahorse XF Analyzers (Agilent, Santa Clara, CA, USA), are estimated by glycolysis and mitochondrial oxidative phosphorylation based on extracellular acidification rate (ECAR) and oxygen consumption rate (OCR), respectively [92]. Bone-marrow derived macrophages at 1 × 10^6^ cells/well were stimulated, as described above, in Seahorse cell culture plates before replacing by Seahorse media (DMEM complemented with glucose, pyruvate, and L-glutamine) (Agilent, 103575-100) in pH 7.4 at 37 °C for 1 h prior to the challenge with different metabolic interference compounds of mitochondrial and glycolysis stress tests. For the mitochondrial stress test, the samples in Seahorse media (Agilent, 103575-100) were sequentially incubated by oligomycin, carbonyl cyanide 4-(trifluoromethoxy) phenylhydrazone (FCCP), and rotenone/antimycin A. In parallel, glycolysis stress tests were performed in samples with Seahorse XF DMEM medium supplemented with 2 mM Seahorse XF L glutamine. Then, several agents, including glucose, oligomycin, and 2-deoxy-D-glucose (2-DG) were sequentially added. The results in the Seahorse analysis were normalized by applying the total protein abundance in the wave program before measurement of all parameters.

### 4.7. Quantitative Proteomic Analysis

Stimulated macrophages were processed for proteomic analysis as previously described [93]. Briefly, the peptides from LPS/LPS or single LPS macrophages were labeled with 131 and 130 TMT reagents, respectively, dried in vacuum centrifugation, separated into 20 fractions, and concatenated into 10 fractions using the Pierce High pH Reversed-Phase Peptide Fractionation Kit (ThermoFisher). Liquid chromatography–mass spectrometry (LC-MS/MS) analysis of samples was performed on an EASY-nLC1000 system coupled to a Q-Exactive Orbitrap Plus mass spectrometer equipped with a nano-electrospray ion source (ThermoFisher). The MS raw data files were searched against a composite database containing the forward and reversed peptide sequences of the Mouse Swiss-Prot Database with a list of common protein contaminants (www.thegpm.org/crap/, accessed on 29 January 2022). The search parameters were set for the following fixed modifications: carbamidomethylation of cysteine (+57.02146 Da), as well as TMT 6 plex of N-termini and lysine (+229.162932 Da). The reporter ion intensity ratios of LPS/LPS vs. LPS were transformed to log2 (LPS/LPS vs. LPS). *p*-values were calculated with Student’s *t*-test based on the triplicate log2 (LPS/LPS vs. LPS) against 0. The proteins with a *p*-value ≤ 0.05 were considered as significantly altered proteins, and these proteins were subjected to the online DAVID Bioinformatics Resources 6.8 to investigate the enriched biological processes.

### 4.8. Statistical Analysis

Mean ± standard error (SE) was used for data presentation. The differences between groups were examined for statistical significance by one-way analysis of variance (ANOVA) followed by Tukey’s analysis of multiple groups comparison. The time-point experiments were analyzed by the repeated measures ANOVA. Survival analysis was determined by a log-rank test. All statistical analyses were performed with SPSS 11.5 software (SPSS, Chicago, IL, USA) and Graph Pad Prism version 7.0 software (La Jolla, CA, USA). A *p*-value of <0.05 was considered statistically significant.

## Figures and Tables

**Figure 1 ijms-23-01676-f001:**
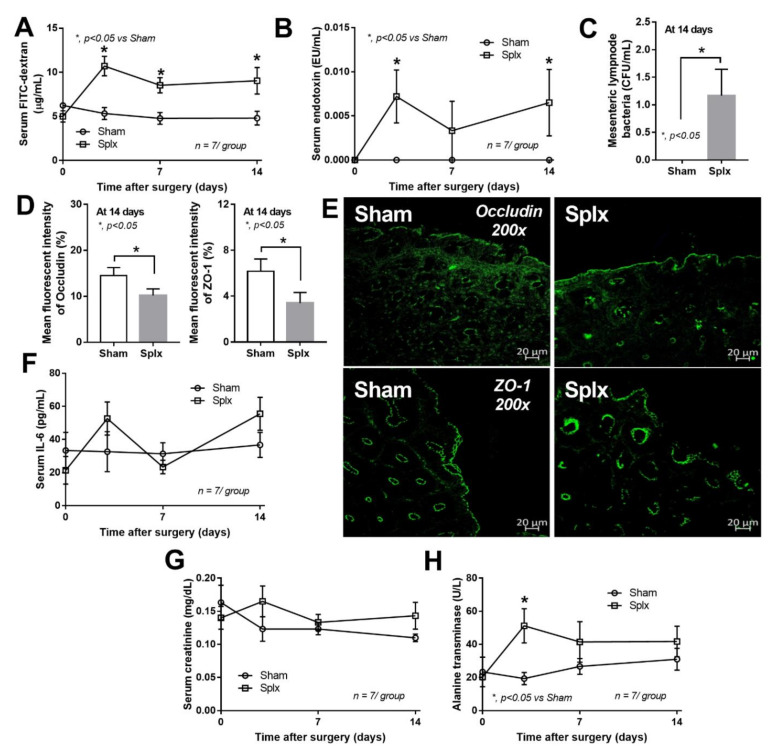
Characteristics of mice after sham or splenectomy (Splx) as determined by leaky gut (FITC-dextran assay and endotoxemia) (**A**,**B**), bacterial burdens in mesenteric lymph nodes, and mean fluorescent intensity of intestinal tight-junction molecules, occludin and zona occludens-1 (ZO-1), with the representative immune-fluorescent staining pictures (**C**–**E**), systemic inflammation (serum IL-6), and organ injury markers (serum creatinine and alanine transaminase) (**F**–**H**) are indicated (*n* = 7/group or time-point).

**Figure 2 ijms-23-01676-f002:**
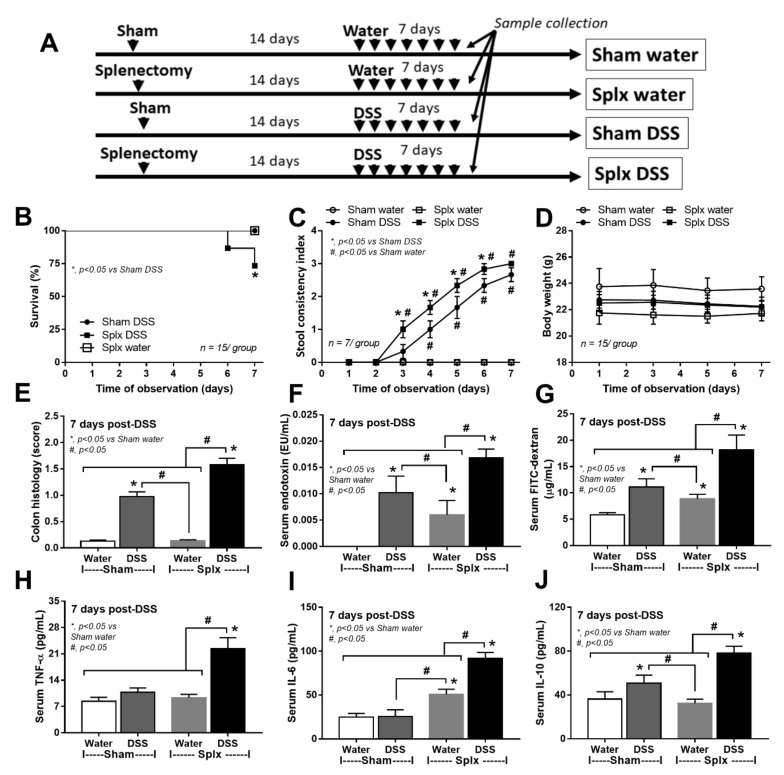
Characteristics of sham or splenectomy (Splx) mice (at 2 weeks post-surgery) after 1 week administration by dextran sulfate solution (DSS) or control drinking water (water) as indicated by the schema of experiments (**A**), survival analysis (**B**), stool consistency index (**C**), body weight (**D**), colon histological score (**E**), gut barrier defect (endotoxemia and FIT-dextran assay) (**F**,**G**), and serum cytokines (TNF-α, IL-6, and IL-10) (**H**–**J**) are demonstrated (*n* = 15/group for B-D and *n* = 8/group for others).

**Figure 3 ijms-23-01676-f003:**
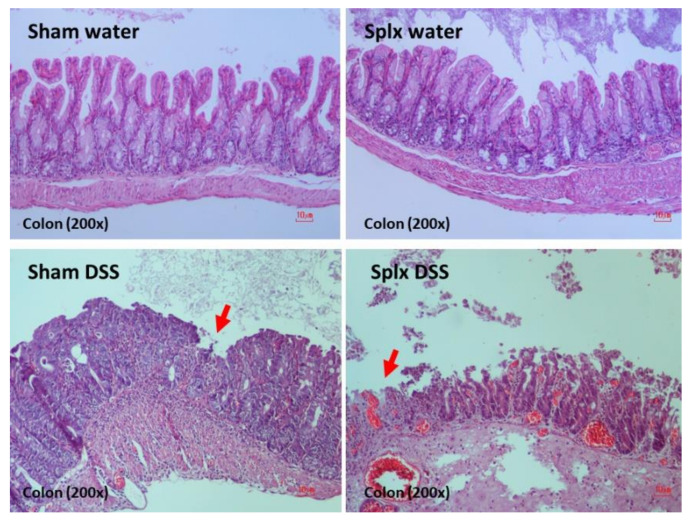
Representative pictures of colon histology from hematoxylin and eosin (H&E) staining of sham or splenectomy (Splx) mice (at 2 weeks post-surgery) after 1 week administration by dextran sulfate solution (DSS) or control drinking water (water) are demonstrated. Arrows indicate the loss of mucosal epithelium in the lesions.

**Figure 4 ijms-23-01676-f004:**
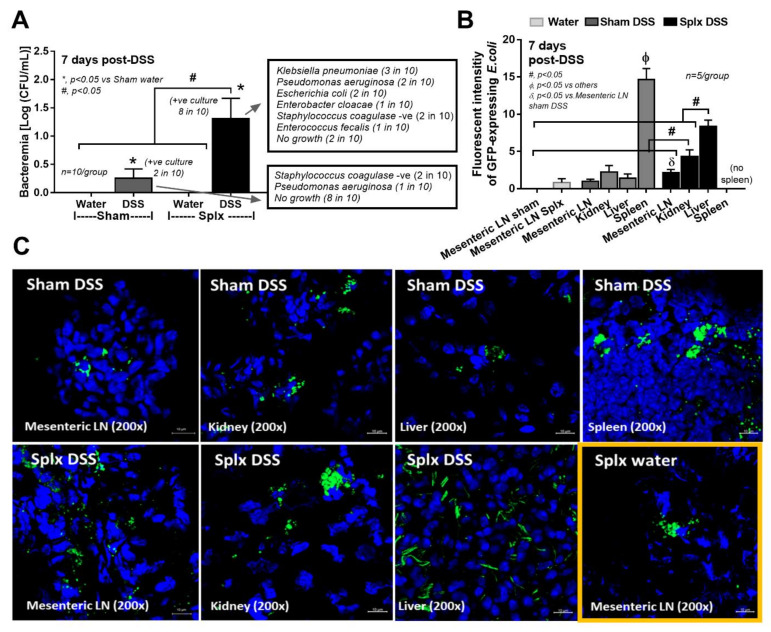
Characteristics of sham or splenectomy (Splx) mice (at 2 weeks post-surgery) after 1 week administration by dextran sulfate solution (DSS) or control drinking water (water) as indicated by blood bacterial burdens (bacteremia) and the list of identified bacteria based on bacterial colony characteristics (mass spectrometry analysis) (**A**), and the fluorescent intensity of green fluorescent protein (GFP)-expressing *Escherichia coli* (*E. coli*) in several organs (mesenteric lymph nodes, kidneys, livers, and spleens) with representative pictures (**B**,**C**) are demonstrated (*n* = 10/group for A, B, and *n* = 5/group for C). Of note, GFP *E. coli* were detectable only in mesenteric lymph nodes, but not other organs, of splenectomy mice with drinking water (Splx water) and non-detectable in all organs of sham mice with drinking water control (sham water).

**Figure 5 ijms-23-01676-f005:**
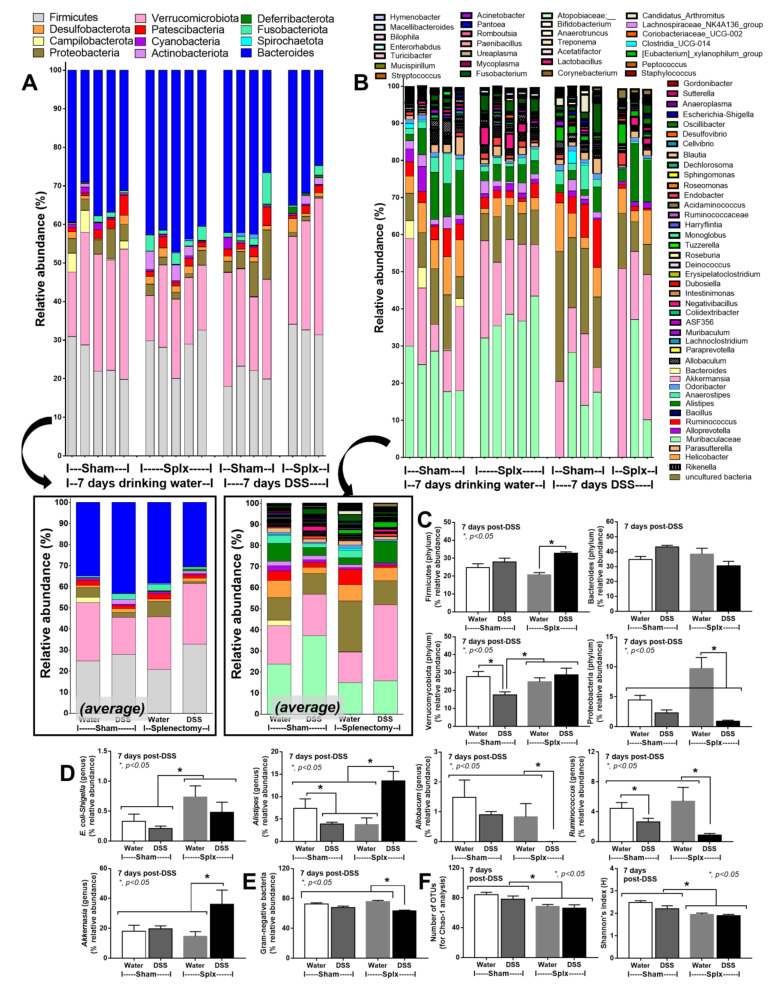
Fecal microbiota analysis of sham or splenectomy (Splx) mice (at 2 weeks post-surgery) after 1 week administration by dextran sulfate solution (DSS) or control drinking water (water) as indicated by the relative abundance of bacterial diversity at phylum and genus with the average calculation (**A**,**B**), the graph presentation of some analysis at phylum and at the genus level (**C**,**D**), total Gram-negative bacteria in feces (**E**), and the beta diversity analysis (Chao-1 and Shannon analysis) (**F**) are demonstrated.

**Figure 6 ijms-23-01676-f006:**
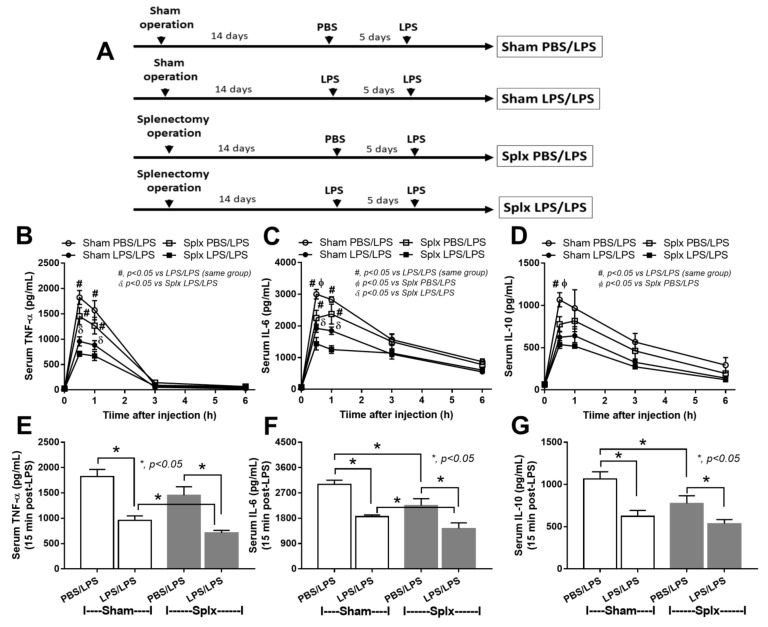
Characteristics of sham or splenectomy (Splx) mice (at 2 weeks post-surgery) with lipopolysaccharide (LPS) intravenous administration (tail vein) in a single dose, using phosphate buffer solution (PBS) followed by LPS (PBS/LPS), or two doses (LPS tolerance) (LPS/LPS) as indicated by the schema of experiments (**A**), time-points of serum cytokines (TNF-α, IL-6, and IL-10) (**B**–**D**), and the graph presentation of peak cytokine levels (15 min post-injection) (**E**–**G**) are demonstrated (*n* = 7/group or time-point).

**Figure 7 ijms-23-01676-f007:**
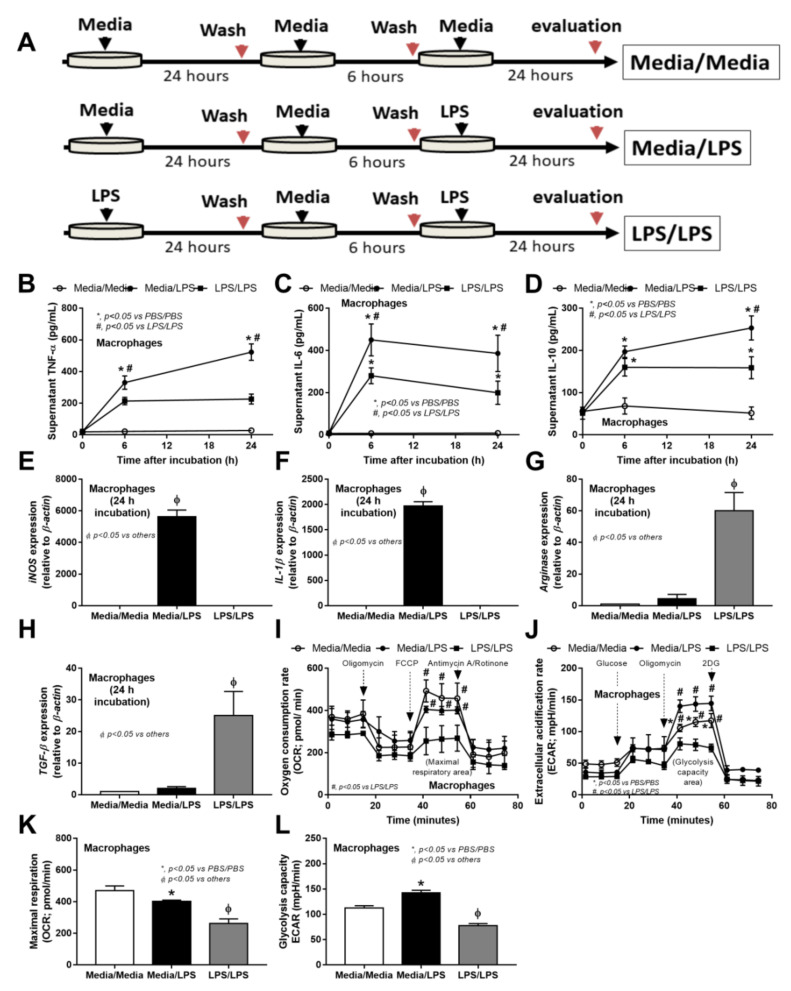
Characteristics of bone marrow-derived macrophages at 24 h after 2-doses stimulation (24 h dose separation) by two doses of phosphate buffer solution (PBS/PBS) (**A**–**L**), a single dose of lipopolysaccharide (LPS), or twice LPS (LPS/LPS) as indicated by the schema of the experiments (**A**), time-points of serum cytokines (TNF-α, IL-6, and IL-10) (**B**–**D**), gene expression of pro-inflammation (*iNOS* and *IL-1β*), and anti-inflammation (*Arginase-1* and *TGF-β*), (**E**–**H**) with cell energy status analysis using oxygen consumption rate (OCR) of the mitochondrial stress test, extracellular acidification rate (ECAR) for the glycolysis stress test with graph presentation of respiratory capacity (maximal respiration), and glycolysis capacity (maximal glycolysis) are demonstrated (independent triplicate experiments were performed).

**Figure 8 ijms-23-01676-f008:**
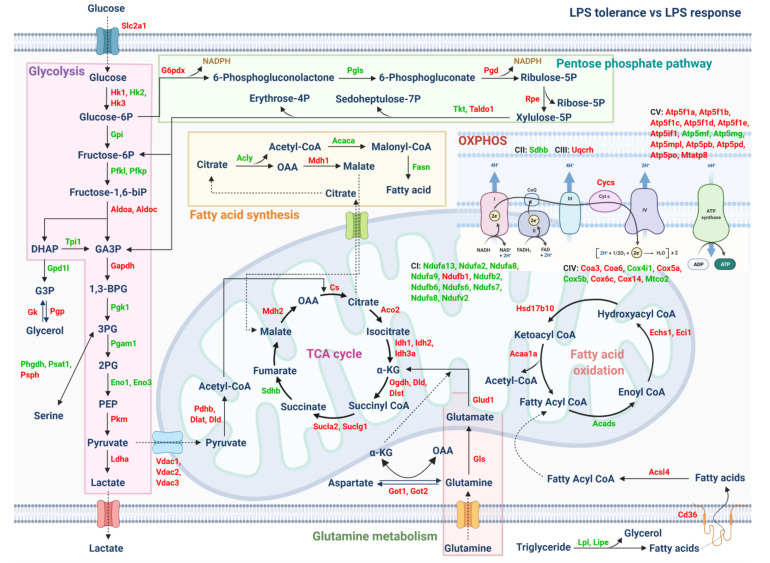
Illustrations of mapped proteins that are associated with cell energy metabolisms from proteomic analysis of bone marrow-derived macrophages after stimulation with two lipopolysaccharide (LPS) administrations (LPS tolerance) relative to single LPS stimulation are demonstrated. The proteins in red- and green-colored texts represent upregulated and downregulated proteins from proteomic analysis, respectively.

**Figure 9 ijms-23-01676-f009:**
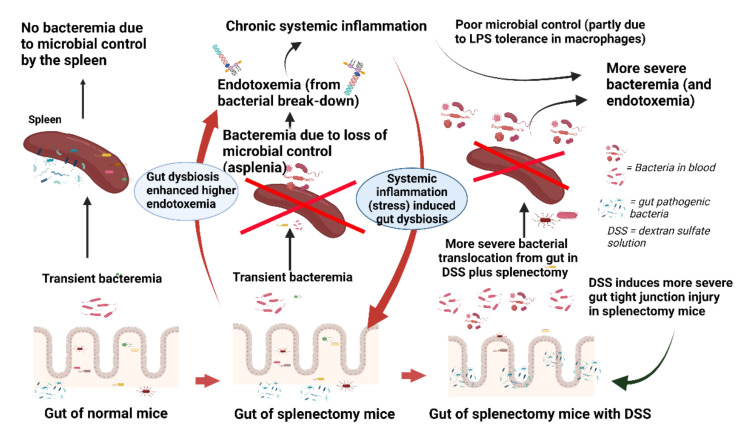
Illustration of the working hypothesis indicating (i) the effective organismal control by spleen after transient bacteremia from physiologic gut barrier defect [45,46], (ii) the loss of microbial control after splenectomy causes minimal gut bacterial translocation but in the level that is enough to control by non-splenic immune responses, as indicated by chronic endotoxemia (splenectomy-induced endotoxemia) without bacteremia [10] (the systemic LPS responses induce stress that additionally facilitates gut dysbiosis [33,47]), (iii) splenectomy-induced gut-dysbiosis enhanced susceptibility to mucositis from dextran sulfate solution (DSS)-induced injury with the higher level of gut bacterial translocation than non-DSS mice (more prominent bacteremia), partly due to the limited cytokine responses from asplenia and chronic endotoxemia-induced macrophage LPS tolerance. Picture is created by https://biorender.com/ (accessed on 10 December 2021).

## Data Availability

The mass spectrometry proteomics data, including annotated spectra for all modified peptides and proteins identified on the basis of a single peptide, have been deposited to the ProteomeXchange Consortium via the PRoteomics IDEntifications (PRIDE) partner repository with the data set identifier PXD027471.

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
