# Peer review of "Enhanced Bacteremia in Dextran Sulfate-Induced Colitis in Splenectomy Mice Correlates with Gut Dysbiosis and LPS Tolerance"

_ijms, 2022, doi:10.3390/ijms23031676_

Round 1

Reviewer 1 Report

  • the manuscript has to be revised by a native English speaking person; currently, English grammar and style are of insufficient quality
  • the complicated and long title of the manuscript has to be changed
  • line 59: inflammatory bowel disease (=> instead of disorder)
  • as leaky gut represents a relevant issue for this manuscript and the incorporated data, the authors should add more information about the term and definition of (pathological) bacterial translocation in the introduction section
  • Is there any explanation why il-10 levels increased in serum but did not show any differences in tissue samples?
  • as the authors delivered a detailed discussion section, references and explanations given within the results section should be dissected and if not mentioned redundantly => only mentioned in the discussion section
  • as proteomic analyses delivered huge amount of data, the authors have to select which part is essential for their work => it seems they try to submerged the reader with information; presented and relevant data should be more focused
  • In its current form the manuscript counts ~7000 words (without abstract and references) => the authors should try to reduced the word count to <6000.
  • discussion section should be more concise and especially, potential links and implications for the human setting should be pointed out more in detail as well as overall data should be more sufficiently integrated in the existing body of evidence. In its current form, the manuscript shows a lot of data and seems a bit enumerative. As the data and the context is definitely relevant, thorough revision holds the potential of an interesting and valuable manuscript

Author Response

The manuscript has to be revised by a native English-speaking person; currently, English grammar and style are of insufficient quality

ANS: We apologize for the unclear presentation. We have our manuscript edited by a native English speaker at the research affair of the University before the submission of the revised manuscript. However, we are willing to seek more help from other source of the English Editing service to reduce language barrier in the scientific community if the quality still does not reach minimal standard of the journal.  

the complicated and long title of the manuscript has to be changed

ANS: We thank the reviewer for the comment and change it into “Enhances bacteremia in dextran sulfate-induced colitis in splenectomy mice correlates with gut dysbiosis and LPS tolerance”.

line 59: inflammatory bowel disease (=> instead of disorder)

ANS: We thank the reviewer for the comment and correct it accordingly.

as leaky gut represents a relevant issue for this manuscript and the incorporated data, the authors should add more information about the term and definition of (pathological) bacterial translocation in the introduction section.

ANS: We thank the reviewer for the comment and add the information accordingly in the introduction as following “As such, epithelial tight junctions hold enterocytes together, forming the first phase of the intrinsic mucosal defense system as a selective physical barrier between the host and gut contents [15]. Increased gut permeability (gut barrier defect or leaky gut) causes augmentation on the translocation of viable pathogens, referred to as “gut translocation or gut bacterial translocation”, or pathogen molecules, including LPS, through the gut wall into blood circulation [16,17].”. 

Is there any explanation why il-10 levels increased in serum but did not show any differences in tissue samples?

ANS: We thank the reviewer for the comment. Due to the comment on the limitation of tissue cytokine that raised by the reviewer 2, we cut all of the data on colon cytokine from the manuscript.

as the authors delivered a detailed discussion section, references and explanations given within the results section should be dissected and if not mentioned redundantly => only mentioned in the discussion section

ANS: We thank the reviewer for the comment and delete them accordingly.

as proteomic analyses delivered huge amount of data, the authors have to select which part is essential for their work => it seems they try to submerged the reader with information; presented and relevant data should be more focused

ANS: We thank the reviewer for the comment. We remove the irrelevant data and keep only proteomic on the cell energy (to relate to extracellular flux analysis).

In its current form the manuscript counts ~7000 words (without abstract and references) => the authors should try to reduce the word count to <6000.

ANS: We thank the reviewer for the comment. Nearly all parts of the manuscript (methods, results, and discussion) were shortening as the manuscript counts ~5000 words (without abstract and references).

discussion section should be more concise and especially, potential links and implications for the human setting should be pointed out more in detail as well as overall data should be more sufficiently integrated in the existing body of evidence. In its current form, the manuscript shows a lot of data and seems a bit enumerative. As the data and the context is definitely relevant, thorough revision holds the potential of an interesting and valuable manuscript

ANS: We thank the reviewer for the comment. We shortened the discussion and briefly pointed out more detail in the human setting in the discussion as following “Precaution for severe infection, treatments for gut dysbiosis, and early antibiotics might be necessary for post-splenectomy individuals with endotoxemia or leaky gut, while immune interventions (enhanced macrophage activities) might be beneficial in these patients with severe infection.”.

Reviewer 2 Report

In general, the manuscript is excellent. But I have some considerations to improve the reading and understanding of the text:

  1. The abstract is not clear. I couldn't understand the objective, importance, and relevance of the study reading it. It must be more compact and "right to the point" in relation to the main results. With an emphasis on the technological methods used.

  1. The first paragraph of the introduction, which presents the problem, could be more concise and objective. Try to focus on the topic addressed in the results. On the other hand, the second paragraph was very superficial. The connection between gut-barrier defect, gut dysbiosis, macrophages, cytokines, immune system imbalance should be more detailed. And, the importance of the microbiota for the balance of the organism and how it connects with the immune system, sepsis, and immune tolerance must be written.

  1. In the first section of materials and methods the timeline of the protocols of colitis, LPS tolerance, and sepsis were a bit confused. My suggestion is to better detail each of the protocols separated according to the timeline and objective of the strategy.

  1. In line 101 the indication of intestinal cytokines figure is not correct. Please fix it.

  1. Most of ELISA's Kits suggested that the assay can be done in serum, plasma, tissue, and other materials. For serum and plasma cytokine detection, the ELISA is an excellent method. However, tissue contains a lot of debris that interferes with the reaction, and a false-positive result is very common. So, for cytokine detection in the colon, I suggest the use of HPLC or flow cytometry.

  1. In line 127. Please detailed what was measured in the colon histology.

  1. Figure 4 A, B and C must be better designed. It's polluted and confusing.

Author Response

In general, the manuscript is excellent. But I have some considerations to improve the reading and understanding of the text:

The abstract is not clear. I couldn't understand the objective, importance, and relevance of the study reading it. It must be more compact and "right to the point" in relation to the main results. With an emphasis on the technological methods used.

ANS: We thank the reviewer for the comment and rewrite nearly all of the abstract.

The first paragraph of the introduction, which presents the problem, could be more concise and objective. Try to focus on the topic addressed in the results. On the other hand, the second paragraph was very superficial. The connection between gut-barrier defect, gut dysbiosis, macrophages, cytokines, immune system imbalance should be more detailed. And, the importance of the microbiota for the balance of the organism and how it connects with the immune system, sepsis, and immune tolerance must be written.

ANS: We thank the reviewer for the comment. We shorten the 1st part and put the objective of the study on this paragraph as following “Our objective was to demonstrate and explore the clinical impact of intestinal barrier defect post-splenectomy that might facilitate spontaneous systemic infection.”. Then, we add the connection of these topics and the immune responses in 2nd and 3rd paragraph.

In the first section of materials and methods the timeline of the protocols of colitis, LPS tolerance, and sepsis were a bit confused. My suggestion is to better detail each of the protocols separated according to the timeline and objective of the strategy.

ANS: We thank the reviewer for the comment and put the sub-heading on the section as “4.1.1. Splenectomy and dextran sulfate solution and 4.1.2. Single or sequential LPS administration in splenectomy mice” and also put a schema in the figure 2.

In line 101 the indication of intestinal cytokines figure is not correct. Please fix it.

ANS: We thank the reviewer for the comment. We have cut this part due to the removal of colon cytokine from the manuscript.

Most of ELISA's Kits suggested that the assay can be done in serum, plasma, tissue, and other materials. For serum and plasma cytokine detection, the ELISA is an excellent method. However, tissue contains a lot of debris that interferes with the reaction, and a false-positive result is very common. So, for cytokine detection in the colon, I suggest the use of HPLC or flow cytometry.

ANS: We thank the reviewer for the comment. Due to the enumerative amount of the data, we cut colon cytokine from the manuscript.

In line 127. Please detailed what was measured in the colon histology.

ANS: We thank the reviewer for the comment and add the information as following “At 7 days post-DSS, the colon histology score, as determined by leukocyte infiltration, enterocyte injury (hyperplasia and vacuolization), and ulceration, was more severe as show in the Revised Manuscript”.

Figure 4 A, B and C must be better designed. It's polluted and confusing.

ANS: We thank the reviewer for the comment and modify the figure.